# Trends in alcohol-related injury admissions in adolescents in Western Australia and England: population-based cohort study

Melissa O'Donnell,[1] Scott Sims,[1] Miriam J Maclean,[1] Arturo Gonzalez-Izquierdo,[2] Ruth Gilbert,[3] Fiona J Stanley[1]

► Prepublication history and additional material are available. To view these files please visit the journal online (http://dx.doi.org/ 10.1136/bmjopen-2016-014913)

## ABSTRACT

**Background** Alcohol-related harm in young people is now a global health priority. We examined trends in hospital admissions for alcohol-related injuries for adolescents in Western Australia (WA) and in England, identified groups most at risk and determined causes of injuries.

**Methods** Annual incidence rates for alcohol-related injury rates were calculated using population-level hospital admissions data for WA and England. We compared trends in different types of alcohol-related injury by age and gender.

**Results** Despite a decrease in the overall rate of injury admissions for people aged 13–17 years in WA, alcohol-related injuries have increased significantly from 1990 to 2009 (from 8 to 12 per 10 000). Conversely, alcohol-related injury rates have declined in England since 2007. In England, self-harm is the most frequently recorded cause of alcohol-related injury. In WA, unintentional injury is most common; however, violence-related harm is increasing for boys and girls.

**Conclusion** Alcohol-related harm of sufficient severity to require hospital admission is increasing among adolescents in WA. Declining trends in England suggest that this trend is not inevitable or irreversible. More needs to be done to address alcohol-related harm, and on-going monitoring is required to assess the effectiveness of strategies.

### Strengths and limitations of this study

► This study investigates trends in adolescent hospital admissions for injuries and the documented causes of these injuries.

► Although trends in alcohol-related admissions has been reported in previous research, this adds to the knowledge of trends in different age groups and causes of injury.

► The limitations of this research is that only individuals admitted for injuries who have an alcohol-related diagnosis will be captured, not when the perpetrator had been affected by alcohol, therefore underestimating rates of injury associated with alcohol use.

► There also could be changes in coding practices which may contribute to changes in trends.

## INTRODUCTION

Substance use in young people is considered a global health priority with young people highlighted as a priority group in the World Health Organisations Global strategy on Alcohol.[1] There have been increasing concerns about alcohol use among young people over the last decade, with young people the age group most likely to consume alcohol at levels of high risk.[2 3 4] Alcohol-related violence is a growing problem among youth,[5] and heavy drinking and intoxication are associated with violent offences.[6 7]

Given that alcohol use at harmful levels is a significant risk factor for violence, strategies to reduce violence need to address alcohol misuse.[8 9] We have previously shown that violence-related injury accounts for almost a quarter of all injury admissions to hospital for adolescents aged 16–17 years.[10] Clinicians need information on the burden of alcohol-related injury admissions and the cause of those injuries in order to set in place services to address harmful drinking and reduce the risk of future injury or other health consequences. In addition, trends in alcohol-related injury admissions and variation in trends between genders can inform policy about high-risk groups. Variation in these patterns between countries contributes to evidence on potential service, policy, commercial or cultural influences on alcohol-related injury.

The aim of this study was to determine trends in alcohol-related injury admissions to hospital among adolescents in Western Australia (WA) and in England and whether these admissions are violence-related. WA and England have similar health systems and have comparable rates of average daily alcohol

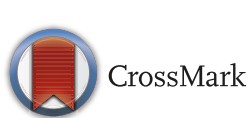

[1]Telethon Kids Institute, University of Western Australia, Perth, Australia
[2]Farr Institute of Health Informatics Research, University College London, London, UK
[3]Institute of Child Health, University College London, London, UK

**Correspondence to**
Dr Melissa O'Donnell;
melissao@ichr.uwa.edu.au

consumption.[11] We also examined whether trends varied by gender and age groups to determine groups most at risk. Our study focused on adolescents aged 13–17 years as this is a priority age group for early intervention and prevention of harmful patterns of alcohol use.[12]

## METHODS
### Data sets
This study used hospital admissions data from England and WA. WA data included adolescents with injury admissions between 13 and 17 years of age to both private and public hospitals between 1990 and 2009. In England, we included the same age group for injury admissions to the National Health Service (NHS) in England between 1997 and 2012. Admissions were excluded if they were planned/scheduled admissions. Therefore, admissions were included if patients were admitted through the hospitals admissions process for a period of care (for any length of time) due to a sudden health issue. In England, each episode of care contains up to 20 diagnostic codes which contain external cause codes and in WA up to 21 diagnostic codes and 4 external cause codes. Both countries use International Classification of Diseases (ICD) coding systems with WA moving from ICD-9 to ICD-10 on 1 July 1999, with England using only ICD-10 from 1997 onwards.

We defined injury admissions based on any diagnostic code for injury, poisoning or other consequences of external cause (ICD Codes: S or T). In WA, ICD-10 and their equivalent codes in ICD-9 were used. Alcohol-related injuries were identified if they contained ICD-10 diagnostic codes which have been used as identifiers for alcohol-related admissions in previous studies:[4 13] F10, E24.4, G31.2, G62.1, G72.1 I42.1, K29.2, K70, K85.2, K86.0, R78.0, T50.6, T51, X45, X65, Y15, Y91, Z50.2, Z71.4, Z72.1.

Causes of injury were classified as intentional if codes were recorded for assault (ICD-10: Y04, Y05, X85-Y03, Y08-Y09) maltreatment (ICD-10: T74, T73, Y06, Y07), self-harm (ICD-10: X60-X69, X70-X84, Z91.5) or undetermined cause (ICD-10: Y20-Y34, Z04.5, Z04.8). The subgroup of violence-related injuries was defined by codes related to assault, maltreatment or undetermined cause which have been used in previous work.[14] Undetermined cause was included as they are an indication of potential assault and self-harm; however, they only make up a small per cent of our intentional injuries (<2%) and alcohol-related injuries (<1%). If individuals had codes in multiple categories (eg, assault and self-harm), they were included in each category. Unintentional injuries were those injury admissions that contained codes of 'accident' as the external cause, including road traffic crashes (V01-V99 and W00-X59).

### Ethics
This study had ethics approval from the University of Western Australia Human Ethics Committee, the Department of Health Human Research Ethics Committee and the Western Australian Aboriginal Human Information and Ethics Committee. Use of anonymised data in England for the purpose in this study did not require research ethics approval but met the requirements of the data provider.[15]

### Analyses
Annual incidence rates for maltreatment-related admissions were calculated by dividing the number of children who had these admissions by age-specific population estimates for each calendar year. In WA, the denominator data were sourced from the Australian Bureau of Statistics population estimates (ABS, 2009). In England, population denominators were based on mid-year population estimates obtained from the UK Office of National Statistics website (UK National Statistics, 2009). Time trends in the annual incidence of admissions were examined using Poisson regression and percentage change per year reported including confidence intervals (CI) to indicate significance. Poisson regression has been recommended for quantifying time trends of discrete events such as injury admissions.[16] All trends are presented as effect sizes with 95% CI and associated p values and were considered statistically significant at the $p < 0.05$ level if the CI did not include zero. A statistically non-significant trend ($p > 0.05$) represents an inconclusive result. Due to some injury categories being relatively rare, there may be an effect which is undetected due to low sample sizes; thus, it is still important to consider effect sizes and CI of trends despite non-significant p values. We compared absolute rates in 2009 as this was the most recent year for data common to both countries. SAS release 9.1 was used to conduct all analyses.[17]

## RESULTS
Injury admissions for any type of injury have declined in both WA and England over the last 10–20 years. However, despite this decrease, WA hospital admissions for alcohol-related injuries for people aged 13–17 years increased during 1990–2009 from 8.2 to 12 per 10 000 people, an increase of 2.3% (95% CI 1.5 to 3.1, p<0.001) per year. Conversely, in England, the rate of admissions for alcohol-related injuries decreased marginally from 9.1 to 6.5 per 10 000 people, over the period of available data from 1997 to 2012, although this was a non-significant decrease (1.1% per year, 95% CI −2.5 to 0.4, p=0.149). Trends in England were relatively stable until 2005 where there were significant decreasing trends for both alcohol-related (−6.2%, 95% CI −9.0 to −3.9, p<0.001) and overall injury rates (−3.1%, 95% CI −4.1 to −2.2, p<0.001), in which alcohol-related injuries had a stronger decline during this period.

Age comparisons indicated that in WA there was an increase in alcohol-related injuries in the 16–17 year age group, an increase of 2.8% (95% CI 1.9 to 3.6, p<0.001) per year (from 14.3 to 22 per 10 000). In the 13–15 year age group there was a non-significant increase of 1.2%

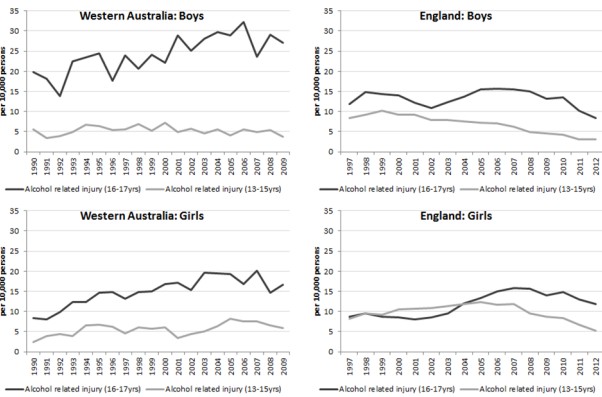

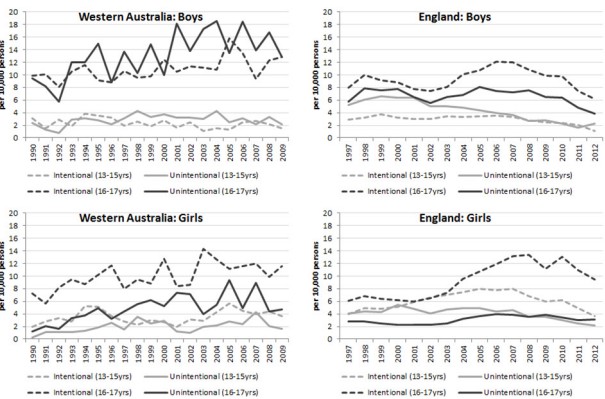

**Figure 1** Rate of hospital admissions for alcohol-related injury in Western Australia and England (per 10 000 persons) for adolescents aged 13–15 and 16–17 years. Light solid=alcohol-related injury (13–15 years); dark solid=alcohol-related injury (16–17 years).

**Figure 2** Rate of intentional and unintentional alcohol-related injury hospital admissions in Western Australia and England (per 10 000 persons) for adolescents aged 13–15 and 16–17 years. Light dash=intentional (13–15 years); light solid=unintentional (13–15 years); dark dash=intentional (16–17 years); dark solid=unintentional (16–17 years).

(95% CI–0.2 to 2.5, p=0.090) per year (from 4 to 4.8 per 10 000). This is in contrast to England where rates for the 16–17 year age group had a non-significant increase of 1.4% (−0.3, 3.1, p=0.101) per year and the rate for 13–15 year age group decreased by 3.7% (−5.4, −1.9, p<0.001) per year.

Gender comparisons showed that WA boys aged 16–17 years had the highest rate of alcohol-related injury admissions at 27 per 10 000 in 2009, whereas the largest change in trends was for girls across both the 13–15 and 16–17 age groups (figure 1). The rates of alcohol-related injury admissions in the 13–15 year age group increased by 2.8% (95% CI 0.9 to 4.6, p<0.010) per year for girls and remained stable for boys (−0.4% per year, 95% CI −1.9 to 1.1, p=0.603). In the 16–17 year age group, girls also had a larger change in trends at 3.3% (95% CI 2.1 to 4.5, p<0.001) per year compared with boys at 2.5% (95% CI 1.4 to 3.5, p<0.001).

In England, there has been an overall increase of 4.1% (95% CI 2.3 to 6.0, p<0.001) per year in alcohol-related injury rates for girls aged 16–17 years (from 9 to 12 per 10 000), whereas rates have declined for all other groups. For boys aged 13–15 years, there has been a significant decrease of 6.5% (95% CI −8.0 to −5.0, p<0.001) per year. For girls aged 13–15 years and boys aged 16–17 years, there have been non-significant decreases of 1.6% (95% CI −3.7 to 0.6, p=0.147) and 0.8% (95% CI −2.5, to 0.9, p=0.345) per year, respectively. Alcohol-related injury rates in England have been decreasing for both genders since 2007, with the most recent rate in 2012 being much lower in England compared with WA for boys aged 16–17 years.

In WA, the predominant cause for alcohol-related injuries for boys aged 13–17 years was unintentional, closely followed by intentional injuries. The opposite was true for girls aged 13–17 years, where intentional injury rates were twice as high as unintentional injury rates. This difference was most notable among girls aged 16–17 years where there were 11.5 per 10 000 intentional alcohol-related

injuries in 2009 compared with 4.8 per 10 000 unintentional alcohol-related injuries (figure 2). In England, the predominant cause of alcohol-related injury was intentional, with girls aged 16–17 years, in particular, showing the highest rates (10.6 per 10 000 in 2009). This was one of the few trends that mirrored the pattern shown in WA.

Alcohol-related injuries that were recorded as intentional are shown according to whether records indicated violence (which were predominantly assaults) or self-harm (figure 3). In WA, alcohol-related injuries caused by violence increased for both genders. This is in contrast to England where violence-related injury rates remained relatively stable. In WA, violence-related injuries for boys aged 13–17 years (4 per 10 000 in 2009) were the most prevalent intentional injuries, with the highest being for boys aged 16–17 years at 8 per 10 000, increasing 4.8% (95% CI 2.8 to 6.7, p<0.001) per year. Girls aged 16–17 years have higher rates (2.7 per 10 000 in 2009) compared with girls aged 13–15 years ; however, they remained stable

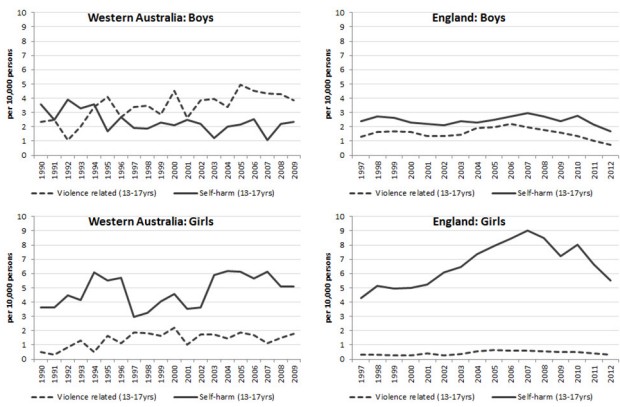

**Figure 3** Rate of intentional alcohol-related injury hospital admissions in Western Australia and England (per 10 000 persons) for adolescents aged 13–17 years, by violence-related and self-harm admissions. Dark dash=violence related (13–17 years); dark solid=self-harm (13–17 years).

from 1990 to 2009 but increased significantly for people aged 13–15 year by 7.1% per year (figure 3). Self-harm predominated among girls in both WA and England, showing increasing rates in both countries, but a decline in England since 2007.

Unintentional alcohol-related injury rates have also increased in WA for both boys and girls between 1990 and 2009 apart from boys aged 13–15 years which remained stable. In England, unintentional alcohol-related injury rates have generally fallen or remained stable across both genders.

Absolute rates of alcohol-related injuries are shown for both countries for 2009, the most recent year in which comparable data were available for WA (table 1). For people aged 13–17 years overall, WA had higher rates of alcohol-related injuries in all categories other than self-harm. In both countries, people aged 16–17 years had the highest rate of alcohol-related injuries; however, the rate was twice as high among boys aged 16–17 years in WA (27.1/10 000) than in England (13.2/10 000). This pattern was consistent across both intentional and unintentional injuries. England had higher rates of alcohol-related self-harm, particularly among girls aged 13–15 years (5.1/10 000) than WA (2.5/10 000).

Further analysis of the alcohol-related injury admission was conducted for WA. Overall, the average length of stay associated with alcohol-related admissions was 2.6 days. For self-harm injuries in boys 24% were alcohol-related compared with 13% of self-harm injuries in girls. For both genders, 15%–17% of assault injuries were alcohol related compared with only 3% of accidental injuries. For people aged 16–17 years, alcohol was associated with 18% of assault-related injuries compared with 12% in people aged 13–15 years. In 2009, this was even higher at 27% of assault-related injuries being associated with alcohol in the 16–17 years age group. The predominant type of alcohol-related assault was assault by bodily force (53%), followed by assault by unspecified means (18%), assault by blunt object (14%) and assault by sharp object (10%). In 3% of alcohol-related assault admissions, the assault was rape.

## DISCUSSION

Despite the decline in hospital admissions in WA from 1990 to 2009 for injury for adolescents aged 13–17 years, there has been an increase in alcohol-related injuries by 2.3% per year. This increase has been largest in the 16–17 years age group with boys aged 16–17 years having the highest rates. However, girls in both the 13–15 and 16–17 age groups had the greatest increase in trends of 2.7% and 3.2% per year, respectively. This substantiates and builds on a previous Australian study showing increases in alcohol-related injury admissions in people aged 16–17 years until 2005 and a large rise in hospital presentations for girls.[4]

In contrast to the increasing rates of alcohol-related injuries in WA, England's rates remained relatively stable during the study period. Most age and gender groups have shown decreasing or stable rates of alcohol-related injuries since 2007, with the exception of girls aged 16–17 years. This suggests that alcohol-related harm in adolescents is a problem that can be reduced. Although coding practices can affect the magnitude of the trend, a decrease in overall alcohol-related harms was found despite an increase in England's use of secondary diagnosis codes[18] and is consistent with surveys of people aged 11–15 years from 2001 to 2011 showing a decrease in frequent drinking.[19] In addition, England's Health Behaviour in School-aged Children Report found from 2002 to 2014, there was a 23%–26% reduction in the proportion of people aged 15 years reporting drinking to excess.[20] Recent data from Public Health England has also found a 13.9% fall in hospital admissions caused by alcohol in England for under 18s which corroborates our findings.[21]

In England, self-harm was the leading cause of alcohol-related injury particularly among girls, an issue which warrants further attention. Self-harm was also the most common form of intentional alcohol-related injury for girls in WA. Research has found many adolescents admitted for injuries with self-harm also had drug or alcohol misuse coded and increased likelihood of having experienced violence, suggesting a need for strategies that address multiple forms of adversity.[22]

A large minority of injuries in WA were unintentional injuries; however, there was a rise in violence-related admissions. Overall, alcohol was associated with 16% of assault-related admissions and this was higher in 2009 for adolescents aged 16–17 years, with 20% of assaults associated with alcohol for girls and 30% for boys. This highlights the growing issue in Australia with the rise in violent injuries related to alcohol and the associated harm that adolescents are experiencing. It is important that this issue be monitored, firstly to know the extent of harm and the burden on the health system of alcohol-related injuries and secondly that as strategies are implemented to reduce this burden we can look at the impact on the rate of admissions.

Unfortunately, our data only capture when the individual admitted for the injury has an alcohol-related diagnosis, not when the perpetrator had been affected by alcohol. Therefore, this would certainly underestimate the rates of injury associated with alcohol use, which is also important in capturing the burden. Research suggests that alcohol use by the perpetrator was a factor in over half of physical and sexual assaults[23] and has established associations with child maltreatment[24 25] and road traffic crashes.[26] Future research could use data from police and corrective services to investigate alcohol-related harm. Also, our data only relate to injuries requiring hospitalisation not those seen in the Emergency Departments; therefore, our injury admissions would relate to more serious injuries and violence. Another limitation is that changes in coding over time may influence apparent trends. Our study shows that at least in WA, there is an increase in recognition (whether due to more rigorous reporting or as a rise in actual incidents) of the influence of alcohol in injury admissions.

**Table 1** Alcohol-related injury admission rates for children in Western Australia and England aged 13–17 years, 2009

| Age group, years | Alcohol-related cause of injury | Western Australia | | | England | | |
|---|---|---|---|---|---|---|---|
| | | Boys, Absolute rate 2009 (per 10000) | Girls, Absolute rate 2009 (per 10000) | Total, Absolute rate 2009 (per 10000) | Boys, Absolute rate 2009 (per 10000) | Girls, Absolute rate 2009 (per 10000) | Total, Absolute rate 2009 (per 10000) |
| 13–15 | Injury | 3.7 (2.3, 5.9) | 6.0 (4.1, 8.8) | 4.8 (3.6, 6.5) | 4.6 (4.2, 5.1) | 8.7 (8.1, 9.3) | 6.6 (6.3, 7.0) |
| | Intentional | 1.5 (0.7, 3.2) | 3.7 (2.3, 6.0) | 2.6 (1.7, 3.9) | 1.8 (1.5, 2.1) | 5.5 (5.0, 6.0) | 3.6 (3.3, 3.9) |
| | Violence | 0.9 (0.3, 2.3) | 1.1 (0.5, 2.8) | 1.0 (0.5, 1.9) | 0.7 (0.5, 0.9) | 0.5 (0.4, 0.7) | 0.6 (0.5, 0.7) |
| | Self-harm | 0.6 (0.2, 2.0) | 2.5 (1.4, 4.6) | 1.6 (0.9, 2.6) | 1.1 (1.0, 1.4) | 5.1 (4.7, 5.6) | 3.1 (2.8, 3.3) |
| | Unintentional | 2.2 (1.2, 4.0) | 1.6 (0.8, 3.4) | 1.9 (1.2, 3.0) | 2.8 (2.5, 3.2) | 3.6 (3.2, 4.0) | 3.2 (2.9, 3.5) |
| 16–17 | Injury | 27.1 (21.9, 33.5) | 16.6 (12.6, 22.0) | 22.0 (18.6, 26.1) | 13.2 (12.4, 14.1) | 14.1 (13.2, 15.0) | 13.6 (13.0, 14.3) |
| | Intentional | 12.9 (9.5, 17.6) | 11.5 (8.3, 16.2) | 12.2 (9.7, 15.4) | 6.9 (6.3, 7.6) | 10.6 (9.8, 11.5) | 8.7 (8.2, 9.2) |
| | Violence | 8.4 (5.7, 12.3) | 2.7 (1.4, 5.4) | 5.6 (4.0, 7.9) | 2.9 (2.5, 3.3) | 0.5 (0.4, 0.7) | 1.7 (1.5, 2.0) |
| | Self-harm | 4.8 (2.9, 8.0) | 8.8 (6.0, 13.0) | 6.8 (5.0, 9.2) | 4.1 (3.7, 4.6) | 10.2 (9.4, 11.0) | 7.1 (6.6, 7.5) |
| | Unintentional | 12.9 (9.5, 17.6) | 4.8 (2.8, 8.0) | 8.9 (6.8, 11.7) | 6.5 (5.9, 7.1) | 3.9 (3.4, 4.4) | 5.2 (4.8, 5.6) |
| 13–17 | Injury | 13.1 (10.8, 15.9) | 10.3 (8.2, 12.9) | 11.7 (10.1, 13.6) | 8.2 (7.8, 8.7) | 10.9 (10.4, 11.5) | 9.5 (9.2, 9.9) |
| | Intentional | 6.1 (4.6, 8.1) | 6.9 (5.2, 9.0) | 6.5 (5.3, 7.9) | 3.9 (3.6, 4.2) | 7.6 (7.2, 8.0) | 5.7 (5.4, 6.0) |
| | Violence | 3.9 (2.7, 5.6) | 1.8 (1.0, 3.1) | 2.9 (2.1, 3.9) | 1.6 (1.4, 1.8) | 0.5 (0.4, 0.6) | 1.1 (1.0, 1.2) |
| | Self-harm | 2.3 (1.5, 3.7) | 5.1 (3.7, 7.0) | 3.7 (2.8, 4.8) | 2.4 (2.2, 2.6) | 7.2 (6.8, 7.7) | 4.7 (4.5, 5.0) |
| | Unintentional | 6.5 (4.9, 8.5) | 2.9 (1.9, 4.4) | 4.7 (3.7, 6.0) | 4.3 (4.0, 4.7) | 3.7 (3.4, 4.0) | 4.0 (3.8, 4.2) |

Using hospital admissions data allow monitoring of trends over time, and country comparisons can generate hypotheses about policies or service factors that may account for differences. This is the first study examining and comparing trends in alcohol-related harm in adolescents in WA and England. Identification of age and gender groups at greatest risk and the types of harm they experience are necessary to inform public health initiatives targeting alcohol-related harm in adolescents. In England, it is unclear which specific policies or practices have influenced the decline, but a strategy for reducing alcohol-related harm has been in place since 2004. An annual tax escalator was implemented to boost duties on all alcohol[27] in 2008, and Challenge 25 was subsequently introduced, requiring identification from people who look younger than 25 years when purchasing alcohol. Researchers have also stated that the economic downturn in England and the rise in alcohol taxation may have contributed to stemming the rise in alcohol-related deaths.[28] English researchers have also noted falls in other adversity-related admissions from 2005 to 2011 and discussed the implementation of programmes (eg, Sure Start and Alcohol Misuse Enforcement Campaign), which may have influenced rates.[29]

Some strategies and programmes are in place in Australia to address youth drinking and alcohol-related harm.[2 3] These appear to be having some positive effects, with adolescents being the only group that has shown a decrease in the proportion of people who drink at risky levels.[3] Nonetheless, our research shows that the rates of alcohol-related injury admissions increased in WA, consistent with findings for Victoria, Australia.[4] Possible explanations for this discordance include the hypothesis that subgroups excluded from surveys (such as homeless or truant youth, those who left school prior to year 12 for school surveys or those that chose not to participate) have had increased drinking or that subgroups have increased excessive and risky drinking.[30]

Australia and England follow a number of the WHO recommendations for reducing alcohol-related harm including maximum blood alcohol levels while driving and restrictions on age, sales to intoxicated people and promotion of alcoholic beverages at activities targeting young people.[11] However, clearly, more needs to be done to reduce the rising rates of alcohol-related harm in WA and in England despite the declining rates. Professor Kevin Fenton, Director of Health and Well-being at Public Health England, has stated that 'while this is promising… current levels of harm caused by alcohol remain unacceptably high, especially within the most deprived communities, who suffer the most from poor health in general'.[31]

Evidence suggests that the optimal package of interventions also includes measures to reduce the affordability and availability of alcohol.[32–34] There is evidence that pricing policy can significantly reduce alcohol consumption[35 36] and alcohol-related hospital admissions[34] and deaths.[37] Minimum unit pricing or volumetric pricing are considered effective and cost-effective solutions.[38] In 2012,

Scotland followed Canada in legislating for minimum alcohol unit pricing.[39] Recent research has found that the majority of young risky drinkers support evidence-based policies including price increases if revenue is used to support treatment and prevention of alcohol problems.[40]

**Acknowledgements**   We thank members of CPRU: Terence Stephenson, Catherine Law, Amanda Edwards, Steve Morris, Helen Roberts, Cathy Street, Russell Viner and Miranda Wolpert. We also acknowledge the partnership of the Western Australian Government Departments of Health, Child Protection, Education, Disability Services, Corrective Services and the Attorney General, which provided data for this project. The views expressed are not necessarily those of the Government Departments that have commissioned or supported this project.

**Contributors**   MOD conceptualised the paper, developed the statistical plan, wrote a preliminary draft and revised the paper. RG contributed to the conceptualisation of the paper, contributed to the draft and revised the paper. SS cleaned and analysed the Western Australian data, contributed to the draft and revised the paper. AG-I cleaned and analysed England data, contributed to the draft and revised the paper. MJM contributed to the interpretation of findings, the drafting of the paper and revised the paper. FJS contributed to the conceptualisation of the paper and revised the paper.

**Funding**   The Policy Research Unit in the Health of Children, Young People and Families (CPRU), which is funded by the England Department of Health Policy Research Programme. This is an independent report commissioned and funded by the Department of Health. We thank members of CPRU: Terence Stephenson, Catherine Law, Amanda Edwards, Steve Morris, Helen Roberts, Cathy Street, Russell Viner and Miranda Wolpert. RG and AG-I were also partly supported by awards establishing the Farr Institute of Health Informatics Research (MR/K006584/1). MOD was supported by a National Health and Medical Research Council Early Career Fellowship (1012439). This research was also supported by an Australian Research Council Linkage Project Grant (LP100200507) and an Australian Research Council Discovery Grant (DP110100967).

**Competing interests**   None declared.

**Ethics approval**   Western Australian Department of Health Human Research Ethics Committee.

**Provenance and peer review**   Not commissioned; externally peer reviewed.

**Data sharing statement**   The data utilised in this paper is owned by our respective Government Departments and therefore would require permissions by these Departments for others to access.

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
