## [Reviewer comments · BMJ Open]

ARTICLE DETAILS

TITLE (PROVISIONAL)	Trends in alcohol-related injury admissions in adolescents in Western Australia and England: Population-based cohort study
AUTHORS	O'Donnell, Melissa; Sims, Scott; Maclean, Miriam; Gonzalez-Izquierdo, Arturo; Gilbert, Ruth; Stanley, Fiona

VERSION 1 - REVIEW

REVIEWER	Mark Stokes Deakin University, Australia
REVIEW RETURNED	24-Nov-2016

GENERAL COMMENTS	I request the authors reconsider their time series data and contrast it to policy events in the UK with respect to alcohol pricing, or be more judicious in their association of policy and outcomes in the discussion (P18, Lines 1-33). I would ask the authors be more judicious in their claims about the association of alcohol and violence in the introduction (P6, line 28-30), where it is stated that "Given the strong links between alcohol use and violence, strategies to reduce violence need to address alcohol use. We have previously shown that violence-related injury ". While alcohol use leads to violence in some, for the majority of teenagers, alcohol use does not result in violence. there is some association, and it is greater than among older persons, but it is not of the level of causation suggested by this introduction. The use of the ICD codes for undetermined cause as indicative of intentional injury is not correct, the cause is undetermined, not intentional. This code should be removed and the analysis repeated. Page 9, line 7, after the word "Poisson", the word "regression" is missing Throughout the results, various non-significant results are referred to as though they were changes. If an event is not significant, it cannot be distinguished from a chance event, and is therefore, not a change. Please remove references to changes that were not changes, as they were not significant (ie: P9, Lines 46-51; P10, lines 28-31; etc).
---

REVIEWER	Simon Marmet Addiction Suisse Research Department Switzerland
REVIEW RETURNED	16-Dec-2016

GENERAL COMMENTS	This is overall a well written papers and I only have some minor comments: General remark: Please consider including information about trends in alcohol consumption in England (there is some information about WA), for example from the HBSC study. The HBSC study for England (http://www.hbscengland.com/wp-content/uploads/2015/10/National-Report-2015.pdf) showed a decrease in alcohol consumption from 2002 to 2014, which may be relevant for the interpretation of your results. Page 7 line 26: I'm not sure whether "unplanned" is the right word here. For example, "poisoning due to alcohol" may very well be a planned "injury". Page 7 line 23: Please add the definition of a hospital admission in your study. Do they need to stay overnight in the hospital in order to count as an hospital admissions? Are ambulant treatments included? Page 8 line 9: How did you handle cases that fit into multiple categories? E.g. a child with a diagnosis for assault and intentional self-poisoning? Did you include them in the assault and self-harm categories or is each person only included once? Page 9 line Did overall injuries decline as strong as alcohol-related injuries in England? In other words, did the percentage of alcohol-related injuries on all injuries go down or up? This may be interesting to add. Do you have an explanation for the decrease of overall injuries? Page 9 line 29 and other mention of increases and decreases: Please mention if an increase is significant, for example by providing the p-value. Page 10 line 12 "for boys" is doubled.
--

VERSION 1 – AUTHOR RESPONSE

Reviewer: 1

Reviewer Name: Mark Stokes

Institution and Country: Deakin University, Australia

4) I request the authors reconsider their time series data and contrast it to policy events in the UK with respect to alcohol pricing, or be more judicious in their association of policy and outcomes in the discussion (P18, Lines 1-33).

As the reviewer has suggested we have been more judicious in our associations of policy and outcomes in the discussion and just referred to other studies that have investigated the association of policy and harm. We have therefore reduced some statements made in this paragraph of the discussion (P18).

5) I would ask the authors be more judicious in their claims about the association of alcohol and violence in the introduction (P6, line 28-30), where it is stated that "Given the strong links between alcohol use and violence, strategies to reduce violence need to address alcohol use. We have previously shown that violence-related injury ". While alcohol use leads to violence in some, for the majority of teenagers, alcohol use does not result in violence. there is some association, and it is greater than among older persons, but it is not of the level of causation suggested by this introduction. The reviewer is correct it is not alcohol per se that causes violence so we have changed the wording to:

'Given alcohol use at harmful levels is a significant risk factor for violence, strategies to reduce

violence need to address alcohol misuse.' (Australian Institute of Criminology, 2009; WHO, 2010).

6) The use of the ICD codes for undetermined cause as indicative of intentional injury is not correct, the cause is undetermined, not intentional. This code should be removed and the analysis repeated. Undetermined intent codes are defined as insufficient information to enable a medical or legal authority to make a distinction between accident, self-harm or assault. We have included these codes as they are an indication of potential assault and self-harm and we have used them in previous research related to victimisation as they are an indication where clinicians concerns have been raised (Gonzalez-Izquierdo et al, 2013, BMC Health Services Research, 13, 260). There is only a relatively small proportion of undetermined cause codes in the data: Undetermined Intent made up 1.7% of intentional injuries (0.8% of overall alcohol related). This has now been added in the methods (page 7).

7) Page 9, line 7, after the word "Poisson", the word "regression" is missing
Changed to "Poisson regression"

8) Throughout the results, various non-significant results are referred to as though they were changes. If an event is not significant, it cannot be distinguished from a chance event, and is therefore, not a change. Please remove references to changes that were not changes, as they were not significant (ie: P9, Lines 46-51; P10, lines 28-31; etc).

Due to both the first and second reviewers comments we have now included p-values in addition to the percentage change and confidence intervals. We have also included a statement in the methods as to why these trends were still reported:

'All trends are presented as effect sizes with 95% CI and associated p-values and considered statistically significant at the $p < 0.05$ level if the confidence interval does not include zero. A statistically non-significant trend ($p = 0.05$ level) represents an inconclusive result. Due to some injury categories being relatively rare, there may be an effect which is undetected due to low sample sizes, thus it is still important to consider effect sizes and confidence intervals of trends despite non-significant p-values.' Page 8.

Reviewer: 2

Reviewer Name: Simon Marmet

Institution and Country: Addiction Suisse, Research Department, Switzerland Please state any competing interests or state 'None declared': none

9) General remark: Please consider including information about trends in alcohol consumption in England (there is some information about WA), for example from the HBSC study. The HBSC study for England (<http://www.hbscengland.com/wp-content/uploads/2015/10/National-Report-2015.pdf>) showed a decrease in alcohol consumption from 2002 to 2014, which may be relevant for the interpretation of your results.

We will have now added a sentence in about the HBSC study in the second paragraph in the discussion: In addition, England's Health Behaviour in School-aged Children Report found from 2002-2014 there was a 23-26% reduction in the proportion of 15 year olds reporting drinking to excess (ref).

10) Page 7 line 26: I'm not sure whether "unplanned" is the right word here. For example, "poisoning due to alcohol" may very well be a planned "injury".
We have now added a further line to clarify in the methods regarding unplanned and planned hospital admissions. 'Admissions were excluded if they were planned/scheduled admissions. Therefore admissions were included if patients were admitted through the hospitals admissions process for a period of care (for any length of time) due to a sudden health issue.' Page 6.

11) Page 7 line 23: Please add the definition of a hospital admission in your study. Do they need to stay overnight in the hospital in order to count as an hospital admissions? Are ambulant treatments included?

As above we included an extra sentence to clarify the admissions included. Page 6.

12) Page 8 line 9: How did you handle cases that fit into multiple categories? E.g. a child with a diagnosis for assault and intentional self-poisoning? Did you include them in the assault and self-harm categories or is each person only included once?

We have clarified in the methods that Individuals with multiple categories were included in each category to enable true estimation of trends. Page 7.

13) Page 9 line Did overall injuries decline as strong as alcohol-related injuries in England? In other words, did the percentage of alcohol-related injuries on all injuries go down or up? This may be interesting to add. Do you have an explanation for the decrease of overall injuries?

We have investigated this further and have added the findings into the first paragraph of the results: 'Trends in England were relatively stable until 2005 onwards where there were significant decreasing trends for both alcohol related (-6.2%, 95%CI:-8.98,-3.94, p<.001) and overall injury rates (-3.1%, 95%CI:-4.11,-2.16, p<.001) in which alcohol-related injuries had a stronger decline during this period.'

Page 9.

Unfortunately we don't have an explanation for why there was a decrease in overall injuries.

14) Page 9 line 29 and other mention of increases and decreases: Please mention if an increase is significant, for example by providing the p-value.

P-values have now be included.

15) Page 10 line 12 "for boys" is doubled.

This has been removed.

VERSION 2 – REVIEW

REVIEWER	Mark Stokes Deakin University., Australia
REVIEW RETURNED	19-Feb-2017

GENERAL COMMENTS	The reviewer completed the checklist but made no further comments.
--

REVIEWER	Simon Marmet Addiction Suisse, Research Department, Switzerland
REVIEW RETURNED	30-Jan-2017

GENERAL COMMENTS	The comments in my earlier review have been given adequate consideration. I have no further comments to the revised manuscripts, except two minor details: page 41 line 21, 23 the p-values have no leading zero, in contrast to the other p-values in the manuscript. page 42 line 2 I would suggest replacing p=0.003 with p<0.01
---

VERSION 2 – AUTHOR RESPONSE

Reviewer: 2

page 41 line 21, 23 the p-values have no leading zero, in contrast to the other p-values in the manuscript. - This has been corrected

page 42 line 2 I would suggest replacing $p=0.003$ with $p<0.01$ -This has been corrected.

We have also updated the contributor statement to provide more details on authors contributions.

We are looking forward to receiving the proofs when they are completed and if there are any further questions please do not hesitate to contact me.